# Validity and Reliability of the Emergency Severity Index in a Spanish Hospital

**DOI:** 10.3390/ijerph16224567

**Published:** 2019-11-18

**Authors:** Luis Miguel Cairós-Ventura, Maria de las Mercedes Novo-Muñoz, José Ángel Rodríguez-Gómez, Ángela María Ortega-Benítez, Elena María Ortega-Barreda, Armando Aguirre-Jaime

**Affiliations:** 1Health Services Management of La Palma, Gerencia de Servicios Sanitarios, 38713 La Palma, Spain; eortegab@ull.edu.es; 2Faculty of Health Sciences, Nursing Section, University of La Laguna, 38200 Tenerife, Spain; mernov@ull.edu.es (M.d.l.M.N.-M.); jarogo@ull.edu.es (J.Á.R.-G.); amortega67@gmail.com (Á.M.O.-B.); 3Health Care Research Institute of the Santa Cruz de Tenerife College of Nursing, Colegio Oficial de Enfermeros de Santa Cruz de Tenerife, 38001 Tenerife, Spain; armagujai@gmail.com; 4Tenerife Primary Care Management, Gerencia de Atención Primaria de Tenerife, 38004 Tenerife, Spain; 5European University of the Canary Islands–Member of Laureate International Universities, Tenerife, 38001 Canary Islands, Spain

**Keywords:** emergency service, hospital, triage, validation studies as topic, validity and reliability of the emergency severity index in a Spanish hospital

## Abstract

Saturation in hospital emergency departments is one of the main safety problems for the patient, which can generate negative consequences for their health. In response to this issue, triage systems are developed to organize the flow of patients in order to allow the most urgent ones to be treated first. The Emergency Severity Index (ESI) is the most used triage system in the USA and it has been implemented in the General Hospital of La Palma since 2010. The objective of this study is the validation of the ESI adapted to our hospital through the study of its degree of reliability, as well as the criterion validity. The sample consisted of 240 randomly selected cases, with proportional representation of emergencies attended in 2015 and their fraction of urgent ones (Levels 1 and 2). Criterion validity was estimated by sensitivity, specificity, and predictive result values. For reliability, the degree of agreement among the nurses was studied by means of the adapted kappa index k_c2_. Criterion validity showed a sensitivity of 89% (85–93%) and a specificity of 97% (94–99%), with a positive predictive value of 68% (62–74%) and a negative predictive value of 99% (98–100%) for the discrimination of urgent cases. The reliability analysis showed a k_c2_ = 0.94 (0.84–0.99) index, a very good agreement according to Landis-Koch criteria. The results of our study have shown adequate validity and reliability in the adaptation and implementation of an ESI triage system suited to the specific conditions of a hospital emergency service in Spain.

## 1. Introduction

A saturated hospital emergency service (HES) can cause negative consequences for the assistance of patients [1]. The most extended organizational and functional response to this issue is the integration of a structured triage system [2,3]. Given this situation, in 2010, the General Hospital of La Palma opted for the application of the most used system in the United States, the Emergency Severity Index (ESI) [4,5], which has proven its reliability and validity [2]. The ESI classifies the patient into five levels of emergency, following an algorithm with four decision points: A—Do they require advanced life support? B—Can they wait? C—How many resources will they need? and D—Are vital signs altered? The ESI stipulates a maximum time of 10 minutes until the start of treatment for patients classified as level 2 and immediate treatment for level 1. For the rest of the levels, it is established that the center itself stipulates the maximum waiting times [2].

Triage systems must demonstrate sufficient validity and reliability [6], and it is recommended that they be checked in the implementation service [7,8] especially when in another language, to avoid errors in important concepts [8]. To determine the ESI criterion validity, a consensual reference standard (gold standard) that identifies the actual level of emergency has not been found. The most used alternative is the use of subrogated variables [9,10] such as length of stay, rate and time of hospitalization, admission to ICU, mortality, consumption of resources, vital signs [11,12,13,14,15,16,17], and emergency transportation [18], by themselves or compared with the estimate that was made in triage. The variables subrogated as proxies to a reference standard have the disadvantage of not being conditioned solely by the level of emergency; thus, the length of stay may be influenced by lack of hospital beds or test results [19], an ineffective treatment, or service saturation [20,21]. Some authors have estimated the criterion validity using “double triage” as a standard, with consecutive and independent classification of the patient by two health professionals; the independent opinion of several emergency physicians has been used on the same patient at discharge. The sensitivity and specificity of the system using the opinion of experts has been set as a gold standard [22,23,24,25]. The criterion validity test of the ESI in the HES of La Palma used a combination of several of these approaches.

The hypothesis proposed as a starting point for this study was that the adaptation of the ESI in the HES of the La Palma hospital had sufficient validity and reliability for a safe classification of the emergency patient. The objective of the study was to verify this extreme.

## 2. Material and Methods

Validation study of the ESI in its employment in the HES of the General Hospital of La Palma, Canary Islands, Spain. The sample was made up of patients who came to the hospital between February 2016 and February 2017 and gave their informed consent, or that of their legal representative, for participation. For these patients, a blind and simultaneous triage was performed by two nurses on their arrival and by two physicians at discharge. The physicians had more than 10 years of experience in emergency care, pursuing their expertise to build our “gold standard”. The minimum experience required of nurses in the emergency department for assessing triage patients was 2 years. Patient assessment by nurses followed the ESI algorithm, while the physicians used all of the information collected from the case at discharge, independently assigning the maximum safe waiting times in the event that the service was not saturated.

The criterion for inclusion of patients in the study was to have the triage form and the second assessment recorded, and the criteria for exclusion were not speaking Spanish, abandoning the service, or revoking informed consent.

Participants were registered by: Age, sex, date and time of onset of symptoms, personal history, reason for consultation, vital signs if required, chronic and administered treatment, consumption and results of diagnostic tests, diagnosis, and destination at discharge. In addition to the interconsultations and techniques used, the level of triage was assigned by each nurse and maximum waiting times were estimated by physicians. Finally, the participants’ length of stay in minutes was measured.

In the estimation of criterion validity, the assessment made by the first nurse was used and contrasted with the reference standard. The reference standard was constructed based on the allocation given by the physicians, including for those patients for which both physicians agreed on the assigned triage level and for those cases in which they deviated by one level; the most restrictive was chosen for being the safest, excluding the cases in which they deviated by more than one level. The frequency of cases with over-triage and sub-triage with respect to the reference pattern as an indicator of quality was also assessed, considering over-triage to be that which had been classified with levels higher than those established by the pattern and sub-triage the one classified below.

The design of the sample was based on its proportional representation of the volume of emergencies handled in 2015 (25,127) and the fraction classified as urgent (7.5%), with the strategy of distributing it in strata per day of the week and shift in order to capture the temporal variability in the influx. A volume of 240 cases chosen in this way gave the study a power of 90% in the estimation of criterion validity parameters and agreements of at least 0.25—considered weak according to Landis-Koch criteria [26] as a criterion of relevance—for bilateral hypothesis testing, at a significance level *p* ≤ 0.05 and with estimation of these parameters at 95% confidence intervals.

## 3. Data Processing

Characteristics of the sample were described by summarizing the nominal variables with the relative frequency of their categories, as well as the ordinal and scale variables that did not follow a normal distribution with median (range).

Criterion validity was estimated by sensitivity, specificity, and predictive result values, contrasting the levels of triage assigned to each patient by the first nurse with those of the reference standard. The predictive values of results were estimated taking into consideration the prevalence of classification of priorities in the service.

Reliability was assessed by the degree of agreement on triage assignments among nurses using Cohen’s kappa inter-rater agreement adapted for the bilateral k_c2_ index (Table 1). This discrepancy weighting system penalizes the disagreements of assignment of more than one level to a greater degree, while the discrepancies of one level are penalized in a lesser way. The maximum penalty is assigned when the discrepancy takes place on the most urgent levels. To interpret the data in the kappa index, the scale used by Landis and Koch [26] was used.

The study was authorized by the Health Services Management of La Palma and the Ethics Committee of the University Hospital of the Canary Islands. The Guideline for Good Clinical Practice and the ethical principles of the Declaration of Helsinki were observed.

All the hypothesis tests were applied bilaterally at a statistical significance level *p* ≤ 0.05, and the calculations were made with the help of the statistical processing package SPSS 21.0© from IBM Co^TM^.

## 4. Results

The sample was constituted of 241 patients with a single case of abandonment, for a useful sample of 240 cases with an average age of 43 (with extremes of 0 and 99) years of age, 45% (*n* = 107) of 15–44 years of age, 22% (*n* = 52) of 45–64 years of age, 12% (*n* = 24) of 75 years of age, 11% (*n* = 26) under 15 years of age, and 10% (*n* = 31) of 65–74 years of age. 56% (*n* = 134) were women.

The reasons stated by those who attended were: Traumatic problems (33%), and gastrointestinal problems (15%). The rest of the reasons were distributed among percentages lower than 8%. Of the patients with recorded constants, 15% had at least one of the altered values.

The adaptation of the ESI to the HES of the General Hospital of La Palma established the maximum safe waiting times for the patient (once classified) shown in Table 2. These times are used as quality indicators in emergency care. The length of stay in the service was less than 3 h in 71% of the cases and 3–6 h in 25% of the cases. Four cases remained more than 12 h. The final destination was home discharge in 92% of cases. A total of 8% were admitted, and four of them went to ICU.

Regarding the triage classifications, 52% of the cases were performed by nurses with more than five years of experience. Nursing assessments at level 5 were somewhat less than half of those performed by physicians. With respect to the physicians’ assessments, the number of cases classified in each level was similar among them. The reference pattern was composed of 1% in level 1, 2% in 2, 27% in 3, 47% in 4, and 23% in 5. The reliability of the reference pattern showed a very good absolute agreement of 76% and 24% in discrepancies of ± 1 level, with a k_c2_ = 0.99 (0.78–1.00) index, according to Landis and Koch criteria [26].

In the classification performed by the first nurse, 7% of cases were levels 1 and 2 (Emergency). According to the reference pattern in those levels, 4% were classified. The criterion validity for this classification showed a sensitivity of 89% (CI95%: 85–93%) and a specificity of 97% (CI95%: 94–99%). Considering a prevalence of 8% for these levels, the predictive value of a positive result was 68% (CI95%: 62–74%) and for a negative result, 99% (CI95%: 98–100%). The sensitivity for the classification in level 1 was 100% (CI95%: 99–100%), in level 2 of 80% (CI95%: 75–85%), and for the grouped levels 3–5 it was 97% (CI95%: 94–100%). The criterion validity parameters for each level of triage are presented in Table 3.

Table 4 shows the joint distributions of triage level classifications assigned by the first nurse and the reference standard. The absolute concordance was 55% (CI95%: 49–61%); the rest presented a discordance of ± 1 level. Inter-rater agreement, according to the adapted kappa (k_c2_), was 0.88 (CI95%: 0.82–0.98), which was very good according to Landis and Koch criteria [26], even for the lower limit of its confidence interval. At the analysis of the cases, 30% were classified as over-triage and 15% as sub-triage.

The reliability analysis shows an absolute agreement of 75% among nurses in the classification of cases, with 25% of disagreement of ± 1 level, similar to the classification agreement among physicians. Table 5 shows the agreement in the ESI classification among nurses with k_c2_ = 0.94 (CI95%: 0.84–0.99), which is very good, even for its lowest possibility, according to the Landis and Koch criteria [26].

## 5. Discussion

The results of the study have shown adequate validity and reliability in the adaptation and implementation of an ESI triage system suited to the typical conditions of a hospital emergency service in Spain. The HES of the Hospital General de La Palma has characteristics similar to other emergency services of Spanish hospitals, with an average stay that ranks below the quality indicators establishing the average stay in an emergency department [27]. This service produces 86% of non-admitted patients out of a total of 305 cases per 1000 inhabitants per year and an emergency pressure of 65% [28].

The start-up of the ESI in the HES of the General Hospital of La Palma was carried out following the recommendations of its implementation manual [2] in addition to the structural modifications of the service and the assignment of nursing staff to make its use viable.

The number of patients classified as level 1 according to the ESI was less than 1%, as is the case in similar studies with ESI or other triage systems [22,25]. The triage levels assigned for levels 3–5 in the sample were the same as for the entire population attended in the service during the study period, though it was slightly higher for levels 1–2. This may be due to the fact that, by pursuing the representativeness of the service burden, patients with emergency levels were intentionally sought to shape the sample, and there may have been over-selection of these cases. 

The search for a reference standard at the moment of estimating the predictive criterion validity of the ESI has been a central point in our study. As an approach to the estimation of the criterion validity of the ESI, subrogated variables have been used, considering that the ESI functions as a screening tool to classify patients [7]. In other studies, the expert physician assessment has been used as a reference standard [23,25] to compare the assessment made by the triage nurse regarding this pattern. This does not stop being an assessment of the agreement between two observers in which the sole medical assessment prevails and is not devoid of subjectivity. In other studies, when the criteria of several experts are used, the allocation has been made by open consensus [24]. In our study, we tried to avoid these biases by using as a reference standard the agreement between two physicians who independently assigned the maximum safe waiting time for the patient with all the information of the case at discharge, resolving cases of disagreement with the selection of the most restrictive level assigned. To confirm the reliability of the reference standard, the agreement between the physicians was assessed.

Our results allow us to affirm that, for the typical environment of an emergency service in a Spanish hospital, the ESI triage system has a good capacity to detect whether patients are an emergency or not, thanks to the high predictive value of negative results. On the other hand, the predictive value of the positive results of the ESI in this type of service supposes that one out of every three patients classified as level 2 were not in reality. At first, this does not negatively affect these patients, but patients being classified as level 2 when they are really level 3 can influence treatment times at levels 3–5.

The predictive value of negative results for each level of ESI triage was very low for cases classified as level 4, while for the other levels it was adequate.

In estimating the reliability of emergency triage systems, some studies use Cohen’s weighted or quadratic kappa inter-rater agreement, adding relevance to borderline cases [25]. In our study, we used a system of weighting discrepancies that accounts for the disagreements in the classification of emergency levels with non-emergency ones, more in accordance with clinical criteria. Thus, in the referred study, the 81% agreement obtained is reduced to 78% due to the sub-triages, according to the system we have used.

Important indicators in an emergency triage system are over-triage and sub-triage because of the possible delay in the treatment that they could incur. The percentages of sub-triage that we have obtained are somewhat higher than those of other studies on the ESI, although cases of sub-triage are observed mainly in levels 3–4, with patients of levels 1–2 being correctly classified or classified as over-triage. The reliability of the ESI in the HES of the General Hospital of La Palma obtained very good matches according to the Landis-Koch criteria [26].

The ESI correctly performs the emergency classification function, but with it, patients over 65 years of age are twice as likely to be classified as sub-triage, a phenomenon similar to that observed with other validation studies in triage systems [1,9,29]. A possible cause of this situation could be the fact that persons over 65 are complicated patients, polymedicated and pluripathological, and are thus difficult to catalog.

Our study is affected by several limitations. The first one is that there was 8% of admission in the sample, compared to the 15% observed in the population for the same period. This difference could be due to the fact that it was more complicated to perform simultaneous triage of patients who arrived at the service by ambulance, and many of those patients ended up being admitted. Another limitation would be the excess of patients classified as levels 1–2 with respect to population behavior, possibly due to the intentional search for more urgent patients to shape the sample for the sake of the representativeness of the service burden, which could have produced an excessive choice of these cases. 

Considering these limitations, in light of our results, we can consider that the ESI triage system has good criterion validity with a high predictive value in its negative results, which gives it a good effectiveness in the screening of emergency patients as well as high reliability. These characteristics allow us to recommend its use in the emergency services of Spanish hospitals.

To confirm our results, studies similar to the one described here are required in emergency services in other Spanish hospitals. As a proposal to improve the ESI, the necessary modifications to improve their classification of patients over 65 years should be studied. Among other factors, pluripathology and polymedication should be considered. This would be an interesting topic for new research.

## Figures and Tables

**Table 1 ijerph-16-04567-t001:** Discrepancy weighting system for the estimation of Cohen’s kappa inter-rater agreement adapted for the bilateral k_c2_ index used to assess the reliability of the Emergency Severity Index (ESI) triage system in the Emergency Service Department of the General Hospital of La Palma.

		Nurse 2				
	Levels	1	2	3	4	5
Nurse 1	1	1	0	0	0	0
	2	0	1	0.85	0	0
	3	0	0.85	1	0.95	0
	4	0	0	0.95	1	0.99
	5	0	0	0	0.99	1

**Table 2 ijerph-16-04567-t002:** Maximum waiting times for the patient (once classified) assigned to the ESI triage system in the Emergency Service Department of the General Hospital of La Palma.

Priority Level	1	2	3	4	5
Wait time (min)	0	0	60	90	180
Re-evaluation time (min)	Continuous	Continuous	30	45	90

**Table 3 ijerph-16-04567-t003:** Validity parameters of predictive criteria for each level of triage obtained with the ESI triage system in the Emergency Service Department of the General Hospital of La Palma between February 2016 and February 2017.

Triage Level	Predictive Value of Positive Result % (CI95%)	Predictive Value of Negative Result % (CI95%)
1	100 (99–100)	100 (99–100)
2	65 (59–71)	98 (96–100)
3	55 (49–61)	85 (80–90)
4	65 (59–71)	55 (49–61)
5	36 (30–42)	91 (86–94)

**Table 4 ijerph-16-04567-t004:** Joint distribution of triage classification performed by a nurse using the ESI system and reference standard offered by two physicians independently on the same 240 patients at discharge between July 2016 and July 2017 in the Emergency Service Department of the General Hospital of La Palma.

Triage Level	1	2	3	4	5
1	4	0	0	0	0
2	0	4	8	0	0
3	0	1	32	22	0
4	0	0	24	77	42
5	0	0	0	11	15

**Table 5 ijerph-16-04567-t005:** Concordance in the classification of the level of triage between two nurses independently using the ESI triage system on 240 patients treated in the Emergency Service Department of the General Hospital of La Palma between February 2016 and February 2017.

Triage Level	1	2	3	4	5
1	4	0	0	0	0
2	0	5	7	0	0
3	0	0	42	13	0
4	0	0	24	109	10
5	0	0	0	7	19

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
