# Peer review of "Validity and Reliability of the Emergency Severity Index in a Spanish Hospital"

_ijerph, 2019, doi:10.3390/ijerph16224567_

Round 1

Reviewer 1 Report

There are a couple of typos.

For example, in the abstract, it should be 0.94(0.84-0.99) rather than 0,94(0,84-0,99).

Another example is on Page 3, in the second paragraph of Section 3 `valuesof.'

The statistical calculation seems reasonable.

My major comment is that how these 240 cases were chosen should be further clarified and justified. The current description is not clear enough.

Do these cases well represent the population? It is possible that these cases might be carefully selected to support the conclusion.

Author Response

Dear reviewer, thank you very much for the considerations. I hope to rectify and clarify them with the following statements.

The current description is not clear enough

The strategy of distributing it in strata per day of week, and shift in order to capture the variability in the influx. The sample was randomly stratified per day of week.

Do these cases well represent the population?

The target population is similar, with respect to the age groups, to the selected sample. (https://www.ine.es/jaxi/Datos.htm?path=/t20/e245/p09/a2011/l0/&file=00002002.px)

It is possible that these cases might be carefully selected to support the conclusion.

It is explained in the discussion section, specifically in the third paragraph

Reviewer 2 Report

Below are a series of comments on the paper: 

The keywords used must be MeSH The introduction, although appropriate, requires a greater description of aspects such as: what results of validity and reliability the ESI has without adaptation, the justification, based on other studies, of the need for said classification model to be adapted to different contexts It has focused solely on the criteria of validity and has not explained the phenomenon and the benefits it brings from the published evidence. In addition, the standards regarding attention times, triage classification and studies that raise subtriage and overtriage must be explained. In abstract indicates that 2015 patients were selected and in the 2016-2017 methodology. This must be corrected. It is indicated in the methodology that the participating physicians had more than 10 years of experience and the nurses 2. This should be justified.

There are some elements of the methodology that need more clarity:

Two groups of patient triage are explained. One formed by two nurses, using the ESI. Is this ESI adapted? How have you adapted? In addition, other triage equipment is included, this formed by doctors, who do not use the ESI. What justification does this group have? Do you want to show that ESI is better than having nothing? What should be very clear, is what preparation has been made of the ESI model, justifying why two groups of analysis are included to establish its validity, taking into account that there is already enough evidence that highlights that this model is useful. How your context differs from those previously validated

The selection of the subjects under study must be explained more concretely. It is spoken in the abstract which has been randomized, but it is not explained in the methodology how this randomization is performed.

The results should clarify:

The average age is 43 with an CI of 0-99 or with extremes of 0 and 99? All relative frequencies must be accompanied by absolute frequencies (n) IC must be modified by CI On page 6-7, the results explain when two physicians compare with the classification made by the nurse. This should have been clearly explained in the methodology.

In the discussion:

Page 6 indicates that the results obtained from subtriage are similar to other studies, but references are not provided. t concludes with the need to make modifications to the ESI in those over 65 but it has not been mentioned or explained so far.

Author Response

The keywords used must be MeSH

http://decs.bvs.br/cgi-bin/wxis1660.exe/decsserver/

Descriptor Inglés:

  Emergency Service, Hospital 
Descriptor Español:   Servicio de Urgencia en Hospital 
Definición Inglés:   Hospital department responsible for the administration and provision of immediate medical or surgical care to the emergency patient.
Descriptor Inglés:   Triage 
Descriptor Español:   Triaje 
Descriptor Portugués:   Triagem 
Sinónimos Inglés:   Triages  
Categoría:   N02.421.297.900
SP8.946.117.226.292.265
Definición Inglés:   The sorting out and classification of patients or casualties to determine priority of need and proper place of treatment. 
Descriptor Inglés:   Validation Studies as Topic 
Descriptor Español:   Estudios de Validación como Asunto 
Descriptor Portugués:   Estudos de Validação como Assunto 
Categoría:   E05.337.925
N05.715.360.335.500
SP5.001.012.038.069
Definición Inglés:   Works on research using processes by which the reliability and relevance of a procedure for a specific purpose are established.

In abstract indicates that 2015 patients were selected and in the 2016-2017 methodology

Emergencies attended in 2015, were taken into account only for the calculation of the sample, with proportional representation of  and their fraction of urgent ones (Levels 1 and 2). The patients were selected in the 2016-2017.

Is this ESI adapted? How have you adapted?

The adaptation consisted of the Spanish translation and allocation of waiting times for each level. These times were taken from another triage system, since the ESi does not stipulate them

Triage equipment is included, this formed by doctors, What justification does this group have?

This group was used to build the gold standard and thus be able to Criterion validity was estimated by sensitivity, specificity and predictive result values, contrasting the levels of triage assigned to each patient by the first nurse with those of the reference standard

Page 6 indicates that the results obtained from subtriage are similar to other studies, but references are not provided

Patients over 65 years of age are twice as likely to be classified as sub-triage, a phenomenon similar to that observed with other validation studies in triage systems9,1.

Page 6 indicates that the results obtained from subtriage are similar to other studies, but references are not provided

As a proposal to improve the ESI, the necessary modifications to improve their classification of patients over 65 years should be studied. Pluripatology and polymedication among other factors should be considered. This would be an interesting topic for new research.

Round 2

Reviewer 2 Report

The authors have clarified only some aspects but have not responded or considered the following comments:

The introduction, although appropriate, requires a greater description of aspects such as: what results of validity and reliability the ESI has without adaptation, the justification, based on other studies, of the need for said classification model to be adapted to different contexts It has focused solely on the criteria of validity and has not explained the phenomenon and the benefits it brings from the published evidence. In addition, the standards regarding attention times, triage classification and studies that raise subtriage and overtriage must be explained. It is indicated in the methodology that the participating physicians had more than 10 years of experience and the nurses 2. This should be justified. Justify why two groups of analysis are included to establish its validity, taking into account that there is already enough evidence that highlights that this model is useful. How your context differs from those previously validated. The average age is 43 with an CI of 0-99 or with extremes of 0 and 99? All relative frequencies must be accompanied by absolute frequencies (n)

All this must be included in the manuscript, not only explained

Author Response

Dear reviewer, I have made the changes suggested in the attached document (in red).
Thank you
